# Combination Treatment of Intense Pulsed Light Therapy and Meibomian Gland Expression for Evaporative Dry Eye

**DOI:** 10.3390/life12071086

**Published:** 2022-07-20

**Authors:** Kai-Ling Peng, Chen-Jui Chiu, Hsin-I Tuan, Yi-Chen Lee, Pyn-Sing Hsu, Jiunn-Liang Chen

**Affiliations:** 1Department of Ophthalmology, Kaohsiung Veterans General Hospital, 386, Ta-Chung 1st Road, Kaohsiung 813, Taiwan; caropkl65@gmail.com (K.-L.P.); cjchiu@vghks.gov.tw (C.-J.C.); hituan@vghks.gov.tw (H.-I.T.); s58901351@gmail.com (Y.-C.L.); pshsu@vghks.gov.tw (P.-S.H.); 2Shu-Zen Junior College of Medicine and Management, Kaohsiung 82144, Taiwan; 3School of Medicine, National Yang Ming Chiao Tung University, Taipei 112, Taiwan

**Keywords:** dry eye, intense pulsed light therapy, meibomian gland, tear

## Abstract

Dry eye disease (DED) is most commonly caused by evaporative subtypes and mainly induced by meibomian gland dysfunction (MGD). Intense pulsed light (IPL) combined with meibomian gland expression (MGX) is a noninvasive treatment for improvement of ocular discomfort symptoms and MGD. In this prospective study between November 2020 and May 2022, the patients with MGD and abnormal meibomian expressibility that met the criteria of both ocular surface disease index (OSDI) ≥ 13 scores and standardized patient evaluation of eye dryness (SPEED) ≥ 8 scores were enrolled in Kaohsiung Veteran General Hospital. Three separate treatment sessions of IPL therapy combined with MGX were administered to the lower lids, with an interval of 28 days. Further tear film assessment included lipid layer thickness (LLT), tear meniscus height (TMH), noninvasive tear break-up time (NIBUT), and meibomian gland loss (MGL) either before or after first and third IPL therapy combined with MGX. In addition, lissamine green staining and pain scores were also recorded. We totally enrolled 37 patients of 74 eyes. Men accounted for 18.92% (7/37). The mean age was 54.51 ± 11.72 years. The mean OSDI scores were 58.12 ± 22, while the SPEED scores were 17.03 ± 5.98. The mean Schirmer’s test was 3.66 ± 2.43 mm. After three sessions of IPL treatment with MGX, the OSDI, SPEED, LLT, TMH, MGL, MGXS, and pain scores were significantly improved. For the MGX scores (MGXS) ≤ 20 group, lissamine green scores showed nearly significant improvements. For the MGXS > 20 group, TMH revealed statistical improvement. Noninvasive IPL therapy with MGX statistically improved not only dry eye symptoms, but also tear film assessments, including LLT, TMH, and MGL.

## 1. Introduction

Dry eye disease (DED) is a multi-factorial ocular surface disease characterized by inadequate or unstable tear film, resulting in disruption of lacrimal homeostasis due to impairment of one or more of its components [1]. Most DED cases are caused by evaporative subtypes, mainly induced by meibomian gland dysfunction (MGD). DED manifests as ocular surface burning and irritation, fluctuating visual acuity, red eye, and epiphora [2].

In patients with MGD, the glands narrow, acini atrophy and hyperkeratosis occurs [3], and meibum viscosity increases [4]. The reduced meibum outflow may encourage proliferation of commensal bacteria [5], which secrete lipases that can change the lipid composition in the meibum, increasing the esterified cholesterol levels and consequently reducing meibomian gland (MG) output [4,6]. Some patients with plugged or capped MG orifices may present with lid margin thickening, irregularity, telangiectasia, and hyperemia [7]. In severe MGD, solidified toothpaste-like secretions can be observed [6,8]. Forced MG expression (MGX), conceptualized in 1921 by Gifford [9], is an effective method for rehabilitating MG and improving dry eye symptoms.

Intense pulsed light (IPL) therapy is widely used in cosmetic skin treatments and for removing hypertrichosis, benign cavernous hemangiomas, benign venous malformation, telangiectasia, port-wine stains, and pigmented lesions [10]. IPL therapy (high-intensity light source consisting of visible light; wavelength 515–1200 nm) postulates that oxyhemoglobin in blood vessels located on the surface of the skin absorbs light emitted from the flash lamp. This absorption generates heat that coagulates red blood cells, leading to blood vessel thrombosis [11,12,13,14]. In addition to reduction in telangiectasia and facial erythema severity, concurrent ocular surface health improvements were observed in patients undergoing IPL for rosacea dermatologic manifestations [15]. This study aimed to assess the performance of combination IPL with MGX in altering tear film characteristics, meibomian expressibility, and improving subjective symptoms associated with *DED* and *MGD.*

## 2. Methods

### 2.1. Patient Recruitment

This prospective study followed the tenets of the Declaration of Helsinki and was approved by our institutional review board. After explaining the informed consent requirements, all enrolled patients provided written consent. In this study, all 37 patients enrolled in Kaohsiung Veteran General Hospital between November 2020 and May 2021 and met the inclusion criteria: MGD with abnormal meibomian expressibility, an ocular surface disease index (OSDI) score ≥ 13, and standardized patient evaluation of eye dryness (SPEED) score ≥ 8. An MGD diagnosis was based on lid margin abnormalities (orifice plugging, lid margin hyperemia, telangiectasia, anterior or posterior shift of the mucocutaneous junction) determined by an experienced ophthalmologist. The exclusion criteria were as follows: patients with contraindications for light therapy (pregnancy, Fitzaptrick skin type 6, sunburn, sunlight allergy, ultraviolet radiation exposure, infectious skin disorders, diabetes, hemophilia, epilepsy, photosensitive therapy, pacemaker, defibrillation, cutaneous purpura, and cutaneous disorders [including acne, birthmarks, and eczema]). Nevus and tattoos should be protected during IPL treatment. Participants who received clinical skin treatment within 2 months before this study were also excluded. Wearing contact lenses within 48 h of commencement or during the study, intraocular surgery within 6 months, intraocular or periocular injection within 6 months, any acute infectious or noninfectious ocular condition in either eye within 30 days, and ocular surface disease or condition associated with clinically significant scarring or destruction of conjunctiva or cornea also resulted in exclusion.

### 2.2. Pretreatment Evaluation

We evaluated several enrolled patients’ characteristics (age, sex, pretreatment vision, OSDI and SPEED questionnaires, and Schirmer test) and tear film assessment (lipid layer thickness (LLT), tear meniscus height (TMH), noninvasive tear break-up time (NIBUT), and MG loss (MGL), including upper (UMGL) and lower lids (LMGL)) using the IDRA ocular surface analyzer (SBM SistemiSrl, Orbassano, Italy), and lissamine green scores. For MGL, we calculated the sum of UMGL and LMGL, which was then divided by two. We stained the lower tarsal conjunctiva of each eye using lissamine green strips under saline drops and took external pictures of each eye in anterior and everted lower and upper tarsus views after waiting five minutes to expose the stained area of the lid margins. We modified the 2010 SICCA-Ocular staining score [16] and graded the nasal and temporal conjunctiva with lissamine green staining as follows: 0–9 staining spots, grade 0; 10–30, grade 1; 30–100, grade 2; and >100, as grade 3. Each staining patch was considered as one point. Subsequently, we graded the eyelid margins with lissamine green staining of the horizontal length and vertical percentage over the upper and lower eyelid margins, respectively, according to the Korb grading system for lid wiper epitheliopathy [17,18]. Lissamine green scores of each eye were graded as the sum of nasal, temporal conjunctiva, and upper and lower lid margins.

### 2.3. Treatment Strategy and Evaluation

Both eyes of patients were assessed over an 84-day period, with IPL treatment applied to the skin area immediately below the lower eyelid during three separate treatment sessions on days 0, 28, and 56, with a 28-day interval between each session. Five pulses were applied to four periocular zones inferior to the eye and one periocular zone temporal to the eye; both eyes were protected by opaque goggles. The five pulses were approximately 12 J/cm^2^ each, based on individual skin appearance, as determined by the Fitzpatrick skin type.

After each of the first three IPL treatments, MGX were applied over the bilateral lower lids by meibomian gland expressor forceps. We skipped the MGX procedure if the patient could not tolerate the discomfort of meibomian gland compression. We further graded and recorded the meibum status using the MGX score (MGXS). We modified the international workshop MGD staging as follows [18]. Dysfunctions were graded as 0–3 according to qualitative changes in expressed meibum: complete gland obstruction, grade 0; toothpaste-pattern meibum, grade 1; turbid meibum with debris, grade 2; and clear meibum, grade 3. Fifteen visible main duct orifices of the bilateral lower lids were assessed on biomicroscopy. We recorded the sum of the 15 orifices’ lower lid MG grades as MGXS. Subsequently, all patients would give pain scores during MG expression from the bilateral lower lids (from 0–10, 0 indicating no pain and 10 indicating severe pain).

OSDI, SPEED, and tear film assessments (LLT, TMH, NIBUT, and MGL) were all assessed and recorded again 28 days after the first and third IPL treatments. The MGXS was calculated as 15 scores if 15 orifices were all toothpaste-patterned, 30 scores if 15 orifices were all cloudy withturbid meibum, and 45 scores if 15 orifices were clear meibum. We further divided patients into two groups according to MGXS after the first IPL treatment. Those with a score ≤ 20 were classified into the MGXS ≤ 20 group as severe MGD; those with a score >20 were classified into the MGXS > 20 group as mild to moderate MGD, while those with a score > 30 were classified as mild MGD. 

### 2.4. Data and Statistical Analysis

We analyzed the relationship between general data and tear film assessment before IPL treatment using OSDI and SPEED scores. Categorical variables were analyzed using independent *t*-test; continuous variables were analyzed using the Pearson correlation test. Comparisons of pre- and post-IPL data were performed at different time points using a paired *t*-test. We further analyzed the general pre-IPL data and improvements from pre-IPL to post-third IPL treatment of the two MGXS groups using independent and paired *t*-test, respectively. Data were analyzed using IBM SPSS software v 20.0 (Armonk, NY, USA). A *p* level < 0.05 was considered significant.

## 3. Results

Ultimately, 37 patients met the inclusion criteria: OSDI score >13 and a SPEED score >8. Men accounted for 18.92% (7/37) of the cohort. The mean age was 54.51 ± 11.72 years (24–76, median: 55). The mean OSDI and SPEED scores were 58.12 ± 22 (18.75–95, median: 59.38) and 17.03 ± 5.98 (8–28, median: 17), respectively. The mean Schirmer’s test result was 3.89 ± 2.81 mm. The mean LLT, TMH, NIBUT, MGL, and lissamine greens scores were 34.78 ± 26.31 nm, 0.18 ± 0.06 s, 4.75 ± 0.98 s, 42.40 ± 16.74 %, and 9.03 ± 4.45 scores, respectively. Table 1 summarizes the baseline information before IPL therapy. Among these data, the factors that correlated with OSDI were sex (*p =* 0.009), SPEED scores (*p =* 0.020), LLT (*p =* 0.012), MGXS (*p =* 0.035), pain scores (*p =* 0.020), and lissamine green scores (*p =* 0.028). The factors that correlated with SPEED score were age (*p =* 0.034), and OSDI (*p =* 0.001) and lissamine green scores (*p =* 0.037). The mean OSDI score of female patients (30/37) was 62.58 ± 20.94; that of male patients (7/30) was 37.3 ± 15.55. 

Table 2 summarizes the OSDI and SPEED score, tear film assessment, MGXS, pain score, and lissamine green scored at pre-IPL and post-first and third IPL therapies. The mean OSDI scores decreased from 58.12 ± 22.16 pre-IPL to 41.19 ± 20.86 and 36.89 ± 18.31 (*p* < 0.001, both) post-first and third IPL therapies, respectively, which was significant. The mean SPEED scores decreased from 17.03 ± 5.93 (pre-IPL) to 13.06 ± 6.96 and 11.53 ± 6.51 (*p* < 0.001, both) after the first and third treatments, respectively, which was statistically significant. The mean LLT increased from 34.74 ± 26.31 nm (pre-IPL) to 51.49 ± 29.17 nm and 53.99 ± 31.19 nm (*p* < 0.001, both); the mean TMH mildly increased from 0.18 ± 0.06 s (pre-IPL) to 0.21 ± 0.07 s (*p =* 0.008) and 0.22 ± 0.14 s (*p =* 0.014); the mean NIBUT mildly increased from 4.75 ± 0.99 s (pre-IPL) to 4.94 ± 1.18 s and 4.88 ± 0.98 s, respectively. These differences were statistically significant. The mean MGL decreased from 41.91 ± 20.30 (pre-IPL) to 32.80 ± 14.17 (*p =* 0.006) and 28.11 ± 11.08 (*p*< 0.001) after the first and third treatments, respectively. The mean MGXS mildly increased from 19.84 ± 6.06 following the first IPL treatment to 23.48 ± 6.42 (*p* < 0.001) following the third IPL treatment; moreover, the mean pain score decreased from 6.18 ± 2.22 (post-first-IPL therapy) to 3.58 ± 1.85 (post-third-IL therapy); this improvement was significant (*p* < 0.001). The mean lissamine green scores mildly decreased from 9.03 ± 4.44 (pre-IPL) to 8.70 ± 3.72 (post-first-IPL therapy). 

Table 3 summarizes the pre-IPL and post-first-IPL general data of the MGXS ≤ 20 and MGXS > 20 groups. Among them, the MGXS ≤ 20 group accounted for 47.30% (35/74), the MGXS > 20 group accounted for 39.19% (24/74), while the MGXS > 30 accounted for just 1.35% (1/74). Five patients received IPL only in the first IPL treatment because they could not tolerate MBX but they completed the combined procedure in the subsequent two treatments. The mean OSDI score was higher in the MGXS ≤ 20 group (64.29 ± 18.05 vs. 54.91 ± 23.55); the mean SPEED scores were similar (17.74 ± 5.11 and 17.38 ± 6.54) between groups. However, the mean LLT was mildly higher in the MGXS ≤ 20 group (34.49 ± 25.55 nm vs. 33.52 ± 26.69 nm). The mean TMH was significantly higher in the MGXS ≤ 20 group (0.19 ± 0.06 mm vs. 0.16 ± 0.06 mm) (*p =* 0.029), while the mean NIBUT was nearly identical in both groups (4.75 ± 0.89 s and 4.75 ± 1.07 s). Nevertheless, the mean MGL was higher in the MGXS ≤ 20 group (34.53 ± 11.08 % vs. 32.38 ± 12.13 %). Furthermore, the mean MGXS was significantly higher (*p* < 0.001) in the MGXS > 20 group (25.55 ± 3.50 scores vs. 15.11 ± 2.69); pain scores were approximately the same in both groups (6.06 ± 2.17 and 6.14 ± 2.42). However, the mean lissamine green scores were higher in the MGXS ≤ 20 group (10.12 ± 4.69 vs. 8.56 ± 4.08).

Table 4 shows the improvements between the first and third IPL therapy treatments combined with MGX in both groups. OSDI (*p =* 0.004/0.002) and SPEED scores (*p =* 0.04/<0.001), LLT (*p =* 0.003/<0.001), MGL (*p =* 0.023/0.005), and pain scores (*p* < 0.001 /<0.001) significantly improved in both groups. The significant improvements observed only in the MGXS ≤ 20 group were increased MGXS (*p* < 0.001) and decreased lissamine green scores (*p =* 0.056), while that observed only in the MGXS > 20 group was increased TMH (*p =* 0.025). However, NIBUT showed a mild increase in both the MGXS ≤ 20 group and the MGXS > 20 group between the first and third IPL therapies combined with MGX. After three sessions’ treatment, there were still 32.43% (24/74) whose MGXS was still smaller than 20 scores, 54.50% (40/74) whose MGXS was between 20 and 30 scores, and 6.76% (5/74) whose MGXS was more than 30 scores. Figure 1 shows the changes in OSDI score, NIBUT, LLT, and TMH after three sessions of IPL–MGX therapy in the MGXS ≤ 20 and >20 groups.

## 4. Discussion

MGD is a chronic, diffuse abnormality of the MG, commonly characterized by terminal duct obstruction and/or qualitative/quantitative changes in glandular secretion [19]. Although MG microstructures can currently be evaluated using in vivo confocal microscopy [20,21], the etiology and pathogenesis of MGD remain unclear.

Warm compresses combined with lubricants are the most common recommended supplementary therapies for MGD-related evaporative dry eye. However, MGD management in clinical practice remains challenging, as patient compliance with physician-recommended self-administered therapies is notoriously poor [22]. IPL therapy is a high-intensity light source which is directed toward the skin tissue and is subsequently absorbed by the targeted structure, resulting in heat production (>80 °C), which destroys pigmented skin lesions. A third-generation IPL device designed specifically for periocular application with multiple homogenously sculpted light pulses has recently become commercially available and is currently the only medically certified IPL device for treating MGD [23].

In our study, the significant collaborative related factors of OSDI were SPEED, LLT, MGXS, and pain scores. Regarding the correlation between OSDI scores and LLT, there was a negative correlation between OSDI and LLT; the higher the OSDI scores, the thinner the LLT. This was possibly due to the fact that dry eye symptom severity increased because of the lower lipid content to protect the tear film from evaporation. Regarding the correlation between OSDI scores and MGXS and pain scores, there was also a negative correlation between OSDI scores and MGXS. However, a positive correlation between OSDI and pain scores was noted. These results indicated that increased dry eye symptom severity was associated with stickier and cloudier meibum and more pain during therapeutic MG expression.

Craig et al. reported that the lipid layer grade and NIBUT significantly improved after three separate sessions of IPL, with four pulses applied for patients with mild to moderate MGD. However, in their prospective, double-masked, paired-eye study, the tear evaporation rate and TMH were not different between treated and control eyes [23]. In our study, LLT, TMH and MGXS significantly improved after three sessions of IPL therapy combined with MGX in both eyes. However, NIBUT was a little longer after treatment, even though LLT and MGXS significantly improved. The mean NIBUT before and after IPL treatment was much shorter than those obtained by Craig et al. [23]. Furthermore, stickier and harder meibum was found during compression in 47.30% (35/74) of patients with MGXS ≤ 20 scores and 39.19% (24/74) of patients with MGXS > 20 scores. The percentage of MGXS ≤ 20 scores decreased to 32.43% (24/74) after three sessions of treatment, while 54.50% (40/74) kept MGXS > 20 scores. This may explain why our patients had severer dry eye symptoms than the patients in the study of Craig et al. study, along with short NIBUT, less aqueous preservation, and severe MGD.

According to Vegunta et al. [24], SPEED scores significantly decreased in 89% of 81 patients, and MG evaluations in 77% of patients significantly increased after four IPL treatments combined with MGX at four-week intervals. Tang et al. [25] further reported that combination IPL–MGX therapy was significantly more effective than warm compresses followed by MGX. In their study, SPEED score was reduced by 38% and 22% in the IPL–MGX and warm compress with MGX groups (*p* < 0.01), respectively, and MG yielding secretion score improved by 197% in the IPL treatment group and 96% in the warm compress with MGX group [25]. In our study, there were significant improvements in the OSDI and SPEED scores, LLT, TMH, MGL, MGXS, and pain scores after the first and the third IPL combined with MGX treatments. All studies showed the same results not only in relieving dry eye symptoms, but in improving lipid conditions, consistent with our results. 

Arita et al. [26] reported a study of refractory meibomian gland dysfunction with 45 patients of 90 eyes who were randomly assigned to receive either IPL–MGX or MGX alone as a control. Each eye underwent eight sessions at 3-week intervals. The IPL–MGX group had significantly improved SPEED scores, 14 to 5.5; LLT, 46 to 66 nm; NIBUT, 2.5 s to 7.0 s; BUT, 2.9 to 6.6 s; and meibum grade, 2.2 to 0.3, from pre-IPL to after the eighth session. Pretreatment mean LLT (46 nm) in the study of Aritae et al. showed still thicker than our total mean data (34.74 ± 26.31 nm) and two individual mean data (34.49 ± 25.55 nm in the MGXS ≤ 20 group and 33.52 ± 26.69 nm in the MGXS > 20 group). NIBUT (2.5 ± 1.2 s) and BUT (2.4 ± 1.2 s) in the study of Arita et al. were obviously lower than our total mean data (4.75 ± 0.99 s) and those (5.28 ± 1.42 s /5.29 ± 1.42 s) observed by Craig et al. [23]. Studies of Craig et al. [23] (from 5.28 ± 1.42 s to 14.11 ± 9.75 s) and Arita et al. [25] (from 2.5 ± 1.2 s to 7.0 ± 2.7 s) showed significant improvements in NIBUT, which differed from our results. Although NIBUT in our study didnot improve significantly after IPL–MGX treatment, the mean NIBUT was slightly longer following the third treatment than pre-IPL; the NIBUT (total: 4.88 ± 0.98 s; the MGXS ≤ 20 group: 5.04 ± 1.11 s) in our study following the third treatment was similar to the third treatment (about 5–6 s) in the study by Arita et al. [24], which was shown in the NIBUT curve after IPL–MGX treatment. Furthermore, we analyzed MGXS and pain scores as MGX. MGXS did improve significantly from the first to third IPL treatment. The pain score after the third session of IPL decreased significantly compared to the pain score after the first session, indicating the IPL treatment helps to soften the meibum within the glands and reduce pain experienced during MGX in subsequent treatments. The easier secretion of a clearer meibum after IPL–MGX therapy significantly thickened LLT and statistically increased TMH. However, final mean LLT (53.99 nm, Figure 1) was not thick enough to cover the whole cornea according to the international grade scale of interferometer IDRA test that LLT beyond 80 nm may appear as a stable and thick lipid layer containing colorful oil [27,28], leading to longer NIBUT. Even total mean lissamine green scores in our study were lower, indicating improvement of ocular surface conditions after first IPL–MGX combined therapy compared to pretreatment values, but the difference was statistically insignificant.

We further divided patients who underwent the first IPL–MGX treatment into two groups by MGXS, which scaled in the study for quantity and near-true meibum quality. We selected 15 orifices of meibum in one eye to represent actual and total meibomian expressibility. Regarding pre-IPL-therapy data, TMH and MGXS were significantly different between the groups. After the third treatment session, OSDI and SPEED scores, LLT, MGL, and pain scores significantly improved in both groups. Furthermore, MGXS and lissamine green scores showed improvement nearly significantly in the MGXS ≤ 20 group, which presented severe MGD. Regarding severe MGD, three sessions of IPL–MGX therapy may improve meibum quality and ocular surface conditions a lot. However, NIBUT and TMH only mildly and insignificantly improved post-treatment. Nevertheless, TMH significantly improved in the MGXS > 20 group, indicating mild to moderate MGD, which has thicker LLT than severe MGD after IPL–MGX therapy but is still not sufficient to cover the whole cornea, either. For mild to moderate MGD, IPL therapy combined with MGX is conducted with the purpose of maintaining more aqueous tear film and mild longer NIBUT.

This study has some limitations. First, we did not choose one eye as a placebo control, which was critical in the study design to reduce risk bias from the patient’s knowledge of which eye had been treated. However, bilateral eye treatment met the actual treatment effect in clinical settings. Second, the skin type of most Taiwanese individuals is classified as Fitzpatrick type 3–4; skin reactivity to light or ultraviolet rays may differ between individuals of other ethnicities. Third, most of our patients were female, which may have reduced the representativeness of our findings. Additionally, this study had a small sample size.

## 5. Conclusions

After three sessions of IPL treatment with MGX, the OSDI, SPEED, LLT, TMH, MGL, MGXS, and pain scores significantly improved compared to pretreatment values. For severe MGD, lissamine green scores have shown nearly significant improvements after IPL therapy. For mild and moderate MGD, TMH was revealed to be statistically improved. Noninvasive IPL therapy with MGX statistically improved dry eye symptoms, as well as tear film stability.

## Figures and Tables

**Figure 1 life-12-01086-f001:**
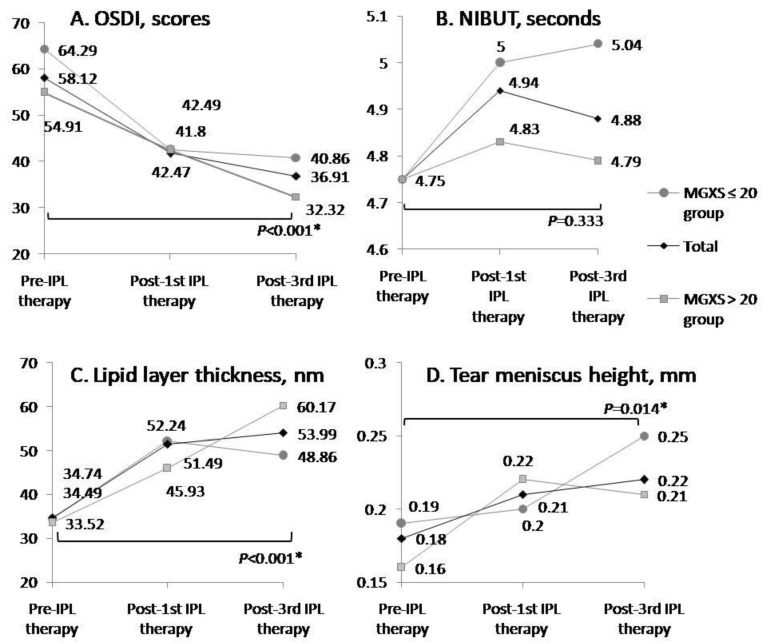
The improvement of mean OSDI scores (total, *p* < 0.001, paired *t*-test), mean lipid layer thickness (LLT) (total, *p* < 0.001, paired *t*-test), and mean tear meniscus height (TMH) (total, *p =* 0.014, paired *t*-test) after three sessions of IPL–MGX combined therapy in either two groups or totally. However, the improvement of mean noninvasive tear break-up time (NIBUT) was not significant (total, *p =* 0.333, paired *t*-test) but the values showed longer times after three sessions of IPL–MGX combined therapy. OSDI scores were overall higher in the MGXS ≤ 20 group (**A**). The mean NIBUT wasgradually improved in these three sessions in the MGXS ≤ 20 group but not inferior to pretreatment in the MGXS > 20 group. (**B**) The improvements of mean LLT werebetter after third treatments in the MGXS > 20 group (**C**). The improvements of mean TMH werebetter after third treatments in the MGXS ≤ 20 group (**D**). * means *p* < 0.05.

**Table 1 life-12-01086-t001:** The baseline information of patients before IPL treatment.

N: 37 Persons (74 Eyes)	N (%), Mean ± SD	OSDI Correlation	SPEED Correlation
Age, years	54.51 ± 11.72	0.530 ^b^	0.034 ^b^*
Sex (M)	7 (18.92%)	0.009 ^a^*	0.097 ^a^
BCVA	0.89 ± 0.25	0.070 ^b^	0.065 ^b^
OSDI, scores	58.12 ± 22.17		0.001 ^b^*
SPEED, scores	17.03 ± 5.98	0.001 ^b^*	
Schirmer test, mm	3.89 ± 2.81	0.728 ^b^	0.125 ^b^
Lipid layer thickness, nm	34.78 ± 26.31	0.012 ^b^*	0.194 ^b^
Tear meniscus height, mm	0.18 ± 0.06	0.051 ^b^	0.281 ^b^
Noninvasive tear break-up time, s	4.75 ± 0.98	0.357 ^b^	0.865 ^b^
Meibomian gland loss, %	42.40 ± 16.74	0.957 ^b^	0.182 ^b^
Meibomian gland expression scores	19.84 ± 6.06	0.035 ^b^*	0.535 ^b^
Pain scores	6.18 ± 2.22	0.020 ^b^*	0.085 ^b^
Lissamine green scores	9.03 ± 4.45	0.028 ^b^*	0.037 ^b^*

* *p* < 0.05, ^a^ independent *t*-test; ^b^ Pearson correlation test; N: number; %: percentage; SD: standarddeviation; M: male; BCVA: best-corrected visual acuity; OSDI: ocular surface disease index; SPEED: standardized patient evaluation of eye dryness; s: seconds.

**Table 2 life-12-01086-t002:** The complete data before IPL and after the first and the third IPL therapies.

N: 37 Persons (74 Eyes)	Pre-IPL Therapy Mean (SD)	Post-1st-IPL Therapy Mean (SD)	*p*	Post-3rd-IPL Therapy Mean (SD)	*p*
OSDI, scores	58.12 (22.16)	41.19 (20.86)	<0.001 *	36.89 (18.31)	<0.001 *
SPEED, scores	17.03 (5.93)	13.06 (6.96)	<0.001 *	11.53 (6.51)	<0.001 *
LLT, nm	34.74 (26.31)	51.49 (29.17)	<0.001 *	53.99 (31.19)	<0.001 *
TMH, mm	0.18 (0.06)	0.21 (0.07)	0.008 *	0.22 (0.14)	0.014 *
NIBUT, s	4.75 (0.99)	4.94 (1.18)	0.233	4.88 (0.98)	0.333
MGL, %	41.91 (20.30)	32.80 (14.17)	0.006 *	28.11 (11.08)	<0.001 *
MGXS, scores		19.84 (6.06)		23.48 (6.42)	<0.001 *
Pain scores		6.18 (2.22)		3.58 (1.85)	<0.001 *
Lissamine green, scores	9.03 (4.44)	8.70 (3.72)	0.576		

* *p* < 0.05, paired *t*-test; N: number; IPL: intense pulse light; SD: standard deviation; OSDI: ocular surface disease index; SPEED: standardized patient evaluation of eye dryness; LLT: lipid layer thickness; TMH: tear meniscus height; NIBUT: noninvasive tear break-up time; s: seconds; %: percentage; MGL: meibomian gland loss; MGXS: meibomian gland expression scores.

**Table 3 life-12-01086-t003:** The pre-IPL and post-first-IPL general data of the MGX scores ≤ 20 and MGX scores > 20 groups.

Pre-IPL Therapy	MGX Scores ≤ 20 (Mean ± SD), N = 35	MGX Scores > 20 (Mean ± SD), N = 29	*p*
Age, years	53.34 ± 13.11	55.62 ± 8.83	0.412
OSDI, scores	64.29 ± 18.05	54.91 ± 23.55	0.094
SPEED, scores	17.74 ± 5.11	17.38 ± 6.54	0.811
LLT, nm	34.49 ± 25.55	33.52 ± 26.69	0.883
TMH, mm	0.19 ± 0.06	0.16 ± 0.06	0.029 *
NIBUT, s	4.75 ± 0.89	4.75 ± 1.07	0.987
MGL, %	34.53 ± 11.08	32.38 ± 12.13	0.584
MGXS, scores	15.11 ± 2.69	25.55 ± 3.50	<0.001 *
Pain scores, scores	6.06 ± 2.17	6.14 ± 2.42	0.892
Lissamine green, scores	10.12 ± 4.69	8.56 ± 4.08	0.190

* *p* < 0.05, independent *t*-test; IPL: intense pulse light; SD: standard deviation; N: number; OSDI: ocular surface disease index; SPEED: standardized patient evaluation of eye dryness; LLT: lipid layer thickness; TMH: tear meniscus height; NIBUT: noninvasive tear break-up time; s: seconds; %: percentage; MGL: meibomian gland loss; MGX: meibomian gland expression; MGXS: meibomian gland expression scores.

**Table 4 life-12-01086-t004:** The data of post-1st-IPL and post-3rd-IPL therapy in two groups of MGXS ≤ 20 and MGXS > 20.

MGXS ≤ 20, N = 35 MGXS > 20, N = 29	MGXS ≤ 20 Pre-IPL Therapy Mean (SD)	MGXS ≤ 20 Post-3rd-IPL Therapy Mean (SD)	MGXS ≤ 20 *p*	MGXS > 20 Pre-IPL Therapy Mean(SD)	MGXS > 20 Post-3rd-IPL Therapy Mean (SD)	MGXS > 20 *p*
OSDI, scores	64.29 (18.05)	40.86 (20.36)	0.004 *	54.91 (23.55)	32.32 (14.99)	0.002 *
SPEED, scores	17.74 (5.11)	13.28 (6.09)	0.019 *	17.38 (6.54)	9.79 (5.42)	<0.001 *
LLT, nm	34.49 (25.55)	48.86 (32.74)	0.003 *	33.52 (26.69)	60.17(28.95)	<0.001 *
TMH, mm	0.19 (0.06)	0.25 (0.18)	0.087	0.16(0.06)	0.21 (0.07)	0.025 *
NIBUT, s	4.75 (0.89)	5.04 (1.11)	0.126	4.75 (1.07)	4.79 (0.82)	0.856
MGL, %	34.53 (11.08)	29.57 (8.00)	0.023 *	43.23 (16.13)	28.56 (10.78)	0.005 *
MGXS, scores	15.11 (2.69)	20.43 (6.04)	<0.001 *	25.55 (3.50)	27.17 (4.76)	0.074
Pain scores	6.06 (2.17)	3.81 (1.79)	<0.001 *	6.14 (2.41)	3.21 (1.88)	<0.001 *
Lissamine green, score	10.12 (4.69)	8.57 (3.47)	0.056	8.56 (4.08)	8.92 (3.66)	0.859

* *p* < 0.05, paired *t*-test; N: number; MGXS: meibomian gland expression scores; IPL: intense pulse light; SD: standard deviation; OSDI: ocular surface disease index; SPEED: standardized patient evaluation of eye dryness; LLT: lipid layer thickness; TMH: tear meniscus height; NIBUT: noninvasive tear break-up time; s: seconds; %: percentage; MGL: meibomian gland loss; MGX: meibomian gland expression.

## Data Availability

The datasets used and/or analyzed during the current study are available from the corresponding author, Jiunn-Liang Chen, on reasonable request.

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
