# Peer review of "Combination Treatment of Intense Pulsed Light Therapy and Meibomian Gland Expression for Evaporative Dry Eye"

_life, 2022, doi:10.3390/life12071086_

Round 1

Reviewer 1 Report

Dear Authors, this is an interesting topic. 

Line 47: "andfor" space needed between words

Line 91: "thestained" space needed between words

Lines 27 to 28: The conclusion section in the abstract section mentions: "Noninvasive IPL therapy with MGX statistically improved not only dry eye symptoms but also tear film assessments". Please could you be more specific as tear film assessments are many and some are more important than others. 

Line 45: Forced MG expression (MGX) the methodology and tools used has not been mentioned. 

Line 226: While the authors report that NIBUT didn't significantly improve, they have not provided sufficient argument to justify this finding.  

Author Response

Reviewer 1:

  1. Comments and Suggestions for Authors: Dear Authors, this is an interesting topic. 

Answer: We thanked for your interest in our topic.

  1. Line 47: "andfor" space needed between words

Answer: Regarding line 47 which showed line 62 in our numbering of this article, we revised it from “andfor” to “and for”.

  1. Line 91: "thestained" space needed between words

Answer: Regarding line 91 which showed line 109 in our numbering of this article, we revised it from “thestained” to “the stained”.

  1. Lines 27 to 28: The conclusion section in the abstract section mentions: "Noninvasive IPL therapy with MGX statistically improved not only dry eye symptoms but also tear film assessments". Please could you be more specific as tear film assessments are many and some are more important than others. 

Answer: Regarding line 27 and 28 which showed line 40 in our numbering of this article, we added “ including LLT, TMH and MGL” to point out the improved items.

  1. Line 45: Forced MG expression (MGX) the methodology and tools used has not been mentioned. 

Answer: Regarding Line 45 which showed line 127 in our numbering of this article, we mentioned about the methodology that after each of the first three IPL treatments, MGX were applied over the bilateral lower lids. We added “by meibomian gland expressor forceps” to explain about the tool. 

  1. Line 226: While the authors report that NIBUT didn't significantly improve, they have not provided sufficient argument to justify this finding.  

Answer: The authors thank the reviewer for the suggestion. Indeed, our findings were identical to other important large-scale studies, except that NIBUT was not statistically significantly improvement but still improved over pretreatment. The results of NIBUT in our study did not contradict those of other studies. We especially focused on comparison our inclusion criteria, methods and results with related information of other reports and tried to explain the possible reasons at the line 287-293, line 319-338 and line 349-359.

A brief explanation is given as follows. The percentage of severe and moderate MGD is higher (47.30% and 39.19% respectively, 8.49% in total) before treatment. After three sessions’ treatments, the percentage reduced to 32.43% in severe MGD and increased to 54.50% in moderate MGD according to MGXS in our study. After treatment of IPL therapy and MGX, mean total MGXS improved from 19.84 ± 6.06 scores to 23.48 ± 6.42 scores while mean pretreatment LLT (34.74 ± 26.31 nm) increased to final mean LLT (53.99 ± 31.19 nm, Figure 1) which was not thick enough to cover the entire cornea. According to the international grade scale of automatic interferometry IDRA test, LLT beyond 80 nm may appear as a stable and thick lipid layer containing colorful oil, leading to longer NIBUT. We added the above content to the section of results (line 200-204, 230-233) and discussions (line 288-290)

We further analyzed the conditions of severe MGD and moderate to mild MGD after three sessions’ treatment. Mean MGXS in the MGXS ≤ 20 group was 15.11 ± 2.69 scores and that in the MGX> 20 group was 25.55 ± 3.50 scores at first while mean MGXS in the MGXS ≤ 20 group improved to 20.43 ± 6.04 scores and in the MGX> 20 group improved to 27.17 ± 4.76 scores after three sessions’ treatment. The improvement of MGXS was significant. Although final mean LLT in the MGXS ≤20 group improved to 48.86 ± 2.69 nm (P<0.001) and in the MGX >20 group improved to 60.17 ± 28.95 nm (P=0.074), both 48.86 ± 2.69 nm and 60.17 ± 28.95 nm were not thick enough to cover the whole cornea, either. Therefore, NIBUT showed a little longer in both groups without significance. The above content we revised the section of discussion at line 311-314.

According to the study of Arita et al. about refractory meibomian gland dysfunction, our pretreatment mean LLT (34.74 ± 26.31 nm) was thinner than their data (46 nm) while our pretreatment mean NIBUT (4.75 ± 0.99 s) was longer than theirs (2.5 ± 1.2 s). However, our final mean LLT (53.99 nm) after three sessions’ treatment was still thinner than theirs (66 nm) after eight sessions’ treatment while our final mean NIBUT (4.88 ± 0.98 s) after three sessions’ treatment was nearly  identical to theirs (5-6 s) after three sessions’ treatment according to their NIBUT curve after IPL-MGX treatment. This may explain that more sessions’ treatment might lead our NIBUT statistically significant improvements. The above content we revised the section of discussion at line 322-325.

Reviewer 2 Report

This manuscript described a study using combination treatment of IPL and MGX in treating dry eye disease. Although I do not have special comments on this manuscript, the author should pay attention to the following details. 

1. What is the statistic method for Fig 1? There is not statistic method mentioned in Fig 1, and also no SD/SEM bar can be found in the figure. 

2. Please do more proof reading. Tables format are not consistent, and also some fonts are different from others (for example, line 190~192). 

Author Response

Reviewer 2:

This manuscript described a study using combination treatment of IPL and MGX in treating dry eye disease. Although I do not have special comments on this manuscript, the author should pay attention to the following details. 

  1. What is the statistic method for Fig 1? There is not statistic method mentioned in Fig 1, and also no SD/SEM bar can be found in the figure. 

Answer: The value in this figure was mean as statistic method. We add “mean” before the items in the figure legends and NIBUT figure. Furthermore, we added mean values near the dots to clear the figures, item units just besides the items and P value of total improvement by paired t-test. We didn’t put the marks of standard deviation because of figure complexity.

We revised the figure legends to” The improvement of mean OSDI scores (total, P<0.001, paired t-test), mean lipid layer thickness (LLT) (total, P<0.001, paired t-test) and mean tear meniscus height (TMH) (total, P=0.014, paired t-test) after three sessions of IPL-MGX combined therapy in either two groups or totally. However, the improvement of mean noninvasive tear breakup time (NIBUT) was not significant (total, P=0.333, paired t-test) but the values showed longer after three sessions of IPL-MGX combined therapy. OSDI scores were overall higher in the MGXS ≤20 group (A). The mean NIBUT showed gradually improved in these three sessions in the MGXS ≤20 group but not inferior to pre-treatment in the MGXS>20 group. (B) The improvements of mean LLT showed better after third treatments in the MGXS>20 group (C). The improvements of mean TMH showed better after third treatments in the MGXS ≤20 group (D).” We added “ NIBUT,” at line 233.

  1. Please do more proof reading. Tables format are not consistent, and also some fonts are different from others (for example, line 190~192). 

Answer: We revised Table 3 about fonts, word spaces, fonts’ size and format.

Pre-IPL therapy

MGX scores ≤20

(Mean ± SD), n=35

MGX scores >20

(Mean ± SD), n=29

P

Age, years

53.34 ±13.11

55.62 ± 8.83

0.412

OSDI, scores

64.29 ± 18.05

54.91 ± 23.55

0.094

SPEED, scores

17.74 ± 5.11

17.38 ± 6.54

0.811

LLT, nm

34.49 ± 25.55

33.52 ± 26.69

0.883

TMH, mm

0.19 ± 0.06

0.16 ± 0.06

0.029*

NIBUT, s

4.75 ± 0.89

4.75 ± 1.07

0.987

MGL, %

34.53 ± 11.08

32.38 ± 12.13

0.584

MGXS, scores

15.11 ± 2.69

25.55 ± 3.50

<0.001*

Pain scores, scores

6.06 ± 2.17

6.14 ± 2.42

0.892

Lissamine green, scores

10.12 ± 4.69

8.56 ± 4.08

0.190

*P<0.05, Independent t-test; IPL: intense pulse light; SD, standard deviation; n: number; OSDI: ocular surface disease index; SPEED: standardized patient evaluation of eye dryness; LLT: lipid layer thickness; TMH: tear meniscus height; NIBUT: non-invasive tear break-up time; s: seconds; %: percentage; MGL: meibomian gland loss; MGX: meibomian gland expression; MGXS: Meibomian gland expression scores

Reviewer 3 Report

Dry eye disease (DED) is considered a complicated and multifactorial disorder of the ocular surface. Several typical clinical symptoms are encompassed by DED, including the sensation of eye dryness, burning, soreness, and ocular pain. In this study, the authors evaluated the efficacy of IPL therapy combined with MGX in DED. They found significantly improved OSDI, SPEED, LLT, TMH, MGL, MGXS and pain scores after the treatments. The study is well-designed, and the results are instructive for clinical practice. A concern is listed below.

1.    Is there any efficacious difference between the MGX scores (MGXS) 20 group and the MGXS >20 group.

Author Response

Answer: The authors thank the reviewer. The international workshop-MGD staging is the most common method for scoring severity of MGD. The authors modified the method as follows. Dysfunctions were graded as 0-3 according to qualitative changes in expressed meibum: complete gland obstruction, grade 0; toothpaste-pattern meibum, grade 1; turbid meibum with debris, grade 2; and clear meibum, grade 3. Fifteen visible main duct orifices of the bilateral lower lids were assessed on biomicroscopy. We recorded the sum of the 15 orifices’ lower lids MG grades as MGXS. The MGXS was calculated 15 scores if 15 orifices were toothpaste pattern, 30 scores if 15 orifices were cloudy and turbid meibum and 45 scores if 15 orifices were clear meibum. We added the above content to the section of method from line 140 to 142.

      We selected patients with an MGXS score of less than or equal to 20 as severe MGD, mainly characterized by toothpaste-like meibum. We selected patients scored more than 20 as moderate MGD, and more than 30 characterized by presence of clear meibum. Because of few eyes with mild MGD in our study, we put the category of mild and moderate MGD together.
